# Is It Time to Redefine the Classification Task for Deep Learning Systems?

Keji Han [1]  Yun Li [1]  Songcan Chen [2]

## Abstract

Many works have demonstrated that deep neural networks (DNNs) are vulnerable to adversarial examples. A deep learning system involves a couple of elements: the learning task, data set, deep model, loss, and optimizer. Each element may cause the vulnerability of the deep learning system, and simply attributing the vulnerability of the deep learning system to the deep model may impede addressing the adversarial attack. So we redefine the robustness of DNNs as the robustness of the deep neural learning system, and we experimentally find that the vulnerability of the deep learning system also roots in the learning task itself. In detail, this paper defines the interval-label classification task for the deep classification system, whose labels are predefined non-overlapping intervals instead of a fixed value (hard label) or probability vector (soft label). The experimental results demonstrate that the interval-label classification task is more robust than the traditional classification task while retaining accuracy.

## 1. Introduction

Deep Neural Networks (DNNs) (Szegedy et al., 2017; Krizhevsky et al., 2017) are even indispensable for many tasks, such as computer vision (CV), natural language processing (NLP) and speech recognition (SR) (LeCun et al., 2015). However, DNNs are demonstrated to be vulnerable to adversarial examples. The adversarial example is crafted by adding adversarial perturbation to the original legitimate example to fool DNNs while being imperceptible to humans.

There are two kinds of adversarial attacks, namely

---

[1]School of Computer Science, Nanjing University of Posts and Telecommunications, Nanjing, P.R.C. [2]Computer Science and Engineering, Nanjing University of Aeronautics and Astronautics, Nanjing, P.R.C.. Correspondence to: Yun Li <liyun@njupt.edu.cn>.

*Accepted by the ICML 2021 workshop on A Blessing in Disguise: The Prospects and Perils of Adversarial Machine Learning.* Copyright 2021 by the author(s).

poisoning attack and evasion attack (Yuan et al., 2019). The poisoning attack ruins the victim model in the training phase, while the evasion attack tries to fool the victim in the testing phase. In this paper, we focus on the evasion attack. The evasion attack falls into three categories according to the manner to craft the adversarial perturbation, namely single-step attack, iterative attack, and optimization-based attack. The single-step attacks directly add the adversarial perturbation into the legitimate example (Goodfellow et al., 2015) or map the legitimate example as an adversarial example(Baluja & Fischer, 2018). The iterative attacks (Kurakin et al., 2017b;a) iteratively explore the adversarial perturbation. For instance, Fast gradient sign method FGSM (Goodfellow et al., 2015) adds the scaled gradient sign to the legitimate example. Project gradient descent (PGD) (Kurakin et al., 2017a) is a multi-step attack, which consists of few single-step attacks. Carlini & Wagner (CW) (Carlini & Wagner, 2017) is also a multi-step attack, which formulates the attack as an optimization problem.

Most of existing methods to improve the robustness of deep learning system pay attention to the training data, the model architecture, training loss, and parameter-updating strategy. For training data-level methods, feature nullification (Wang et al., 2017) randomly nullifies features to improve the robustness of the target model. The work in (Das et al., 2018) demonstrates that image compress can be employed to improve the robustness of DNNs; as to model architecture-level method, in (Xie et al., 2019), the denoising module is introduced to the target model; as to training-loss-based methods, (Zhang et al., 2019; S. et al., 2020) introduce regularization loss; for the parameter-updating strategy methods, (Katz et al., 2017) updates model parameters with the simplex-like method. Jonathan Uesato et al. update model parameters with approximate gradients (Uesato et al., 2018). Moreover, the adversarial example detection methods (Wang et al., 2018; Meng & Chen, 2017; Ma et al., 2019) are also employed to improve the robustness of the deep learning system by keeping the adversarial example away from the deep model.

As introduced above, existing methods mainly focus on elements related to the deep model, and few methods focus on the robustness of the learning task itself. To better investigate adversarial robustness, we extend the robustness

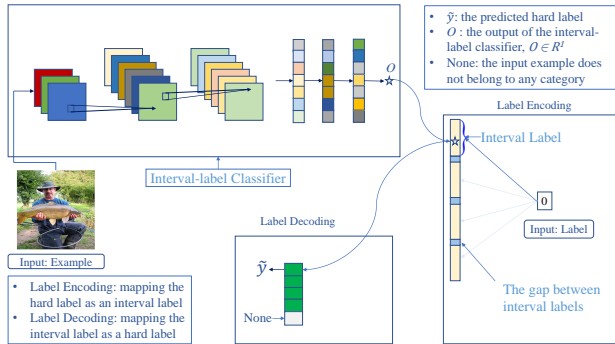

*Figure 1.* The workflow of the interval-label classification system. Unlike the traditional multi-class DNN classifier, the output of the last full-connected layer of the interval-label DNN classifier is a one-dimension scalar. The label of input is the interval corresponding output falls into. 'None' means the input does not belong to any category.

of the deep model to the robustness of the deep learning system. In this paper, we take the deep image classification system as an instance to demonstrate that the vulnerability of deep learning systems may also root in the learning task. Our contributions are summarized as follows.

- We extend the issue of the adversarial robustness of the deep model to the adversarial robustness of the deep learning systems. The deep learning system consists of the predefined learning task and elements to implement the learning task.

- We define the interval-label classification task for the deep learning system. The experimental results demonstrate that it is more robust than the traditional classification task.

## 2. Interval-label Classification

### 2.1. Definition

The traditional image classification task can be defined as follows: $\forall x \in R^{C \times W \times H}, \mathbf{T} : x \mapsto R^d$, where $x$ is an image example, and $C$, $W$ and $H$ are the number of the channels, the width and height of $x$, respectively. $\mathbf{T}$ is the classification task. $k$ is the number of categories. In this paper, we investigate the vulnerability of the classification task by defining the interval-label classification task. The interval-label classification task can defined as: $\forall x \in R^{C \times W \times H}, \mathbf{T} : x \mapsto R^1$. The output of the interval-label classifier, i.e. the deep model to implement the interval-label classification task, is a real number, and the label of $x$ is the predefined interval into which the output falls. The labels of the interval-label classification are some non-overlapping intervals.

### 2.2. An Instance of the Interval-label Classification Task

In this subsection, we introduce an instance of interval-label classification. As introduced in Section 2.1, the interval-label classification maps the example as a real number. So we set the output dimension of the last fully connected layer of the DNN classifier as one. The detailed workflow of the interval-label classification system is shown in Fig. 1. The interval-label classification system consists of three modules, namely label encoding, label decoding module and an interval-label classifier. As mentioned above, the target model is a deep neural network with one-dimension output. The label encoding module turns the hard labels of the existing dataset into predefined interval labels, and the label decoding module projects the interval label as a hard label. When the output of interval-label classifier $o$ does not fall into any label label, Label Decoding module marks corresponding input as 'None'.

Label encoding is vital for the interval-label classification system, which determines the length of each interval label and the gap between two adjacent interval labels. Label Decoding module translates the interval label as a hard label or ground-truth label name, interpreting the classification result to the human. The details of label encoding and label decoding modules will be introduced in the following paragraphs.

Label encoding module transforms the hard label into an interval label. The map function can be formulated as follows,

$$
\begin{aligned}
M_L(y) &= s_0 + y \cdot (\alpha + \beta) \\
M_U(y) &= M_L(y) + \beta
\end{aligned}
\tag{1}
$$

where $s_0$ is the smallest lower bound of interval labels. $\alpha$ is the length of the gap between two adjacent interval labels, while $\beta$ is the length of the interval label. $M_L(\cdot)$ and $M_U(\cdot)$ are lower bound and upper bound map function, respectively. According to Equation (1), for the hard label '3', when $s_0 = 0, \alpha = 1$, and $\beta = 3$, the corresponding interval-label is $[12, 15]$.

If the output of the interval-label classifier falls into an interval label, the label decoding function can be formulated as follows.

$$
\widetilde{y} = \lfloor \frac{\mathcal{I}(x) - s_0}{\alpha + \beta} \rfloor
\tag{2}
$$

where $x$ is the input example. $\mathcal{I}(\cdot)$ is the interval-label classifier. $\lfloor \cdot \rfloor$ is the rounded down function. If $\mathcal{I}(x)$ does not belong to any interval label, the corresponding input example will viewed as an abnormal example. Actually, we can assign variant $\alpha$ and $\beta$ for different interval labels.

The loss function of the interval-label classification task can be formulated as follows,

$$\mathcal{L}(B(X, Y^{'}); \theta) = \|r(\boldsymbol{M_L}(Y) - \mathcal{I}(X)) + r(\mathcal{I}(X) - \boldsymbol{M_U}(Y))\|_2^2 \quad (3)$$

where $\theta$ is the parameter set of the classifier $\mathcal{I}$. $B(X, Y^{'})$ is a mini batch. $X$ and $Y$ is the examples set and hard label set, and $Y^{'} = [\boldsymbol{M_L}(Y), \boldsymbol{M_U}(Y)]$ is the interval labels set. $\boldsymbol{M_L}(Y)$ and $\boldsymbol{M_U}(Y)$ is the lower bound set and upper bound set, respectively. $r(\cdot)$ is the ReLU activation function (Nair & Hinton, 2010).

### 2.3. Characteristics of Interval-label Classification

The traditional classification model more easily overfits compared to the interval-label classification model. The reason is that even an example has been correctly classified by the model, the loss, such as the cross-entropy loss and negative log-likelihood loss, is still not 0 for the traditional classification task. When the probability of the ground-truth class exceeding 0.5, the example certainly is correctly classified. However, only if the target class probability is one that the loss is zero for the cross-entropy loss. As to the interval-label classification task, if an example has been correctly classified, the corresponding loss will be 0, according to Equation (3). According to (Tsipras et al., 2019), overconfidence may cause the adversarial vulnerability of the deep learning system. Moreover, when $s_0 = 0, \alpha = 1$, and $\beta = 0$, the interval-label classification degrades to the traditional classification.

The interval-label classification is immune to existing optimization-based attacks that cannot attack the interval-label classification tasks, such as CW (Carlini & Wagner, 2017), EAD (Chen et al., 2018), and Deepfool (Moosavi-Dezfooli et al., 2016) since the adversary needs to redefine the distance between the output and target label.

## 3. Experiments

### 3.1. Experiment Settings

In this paper, two data sets are applied to evaluate the effectiveness of our method, namely MNIST (Lecun et al., 1998) and a subset of ImageNet (Russakovsky et al., 2015). The training set of MNIST consists of 60,000 grayscale hand-written digit images, while the testing set of MNIST includes 10,000 images. Imagenet is a color image data set, consisting of 1000 fine-grained categories. Considering the task's complexity, we select the first 10 categories of the training set and validation set as the training set and testing set.

On MNIST, three deep neural networks, NN, AlexNet (Krizhevsky et al., 2017) and ResNet18 (He et al., 2016) are employed to implement classification tasks. NN is a neural network consists of two convolution layers and five fully connected layers, and the first two fully connected layers adopt the sigmoid activation function. On subset of ImageNet, AlexNet, DenseNet121 (Huang et al., 2017) and ResNet18 are adopted to implement classification tasks.

### 3.2. Feature Visualization

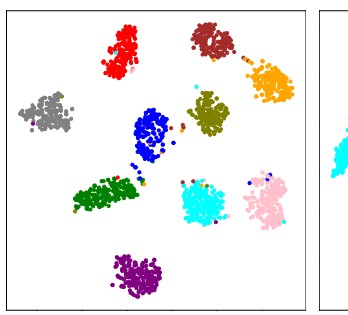 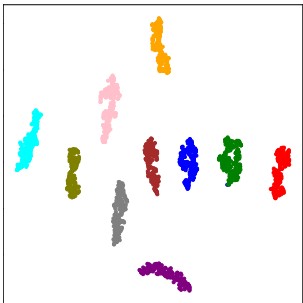

(a) Traditional Classification  (b) Interval-label Classification

*Figure 2.* t-SNE visualization results of the penultimate hidden features of traditional and interval-label classification classifiers on MNIST.

We also experiment to visualize the penultimate layer features for both traditional classification task and interval-label classification on MNIST with t-SNE (Maaten & Geoffrey, 2008). As shown in Fig. 2, for the interval-label classification task, the examples of the same category aggregate more tightly than the traditional classification task. There is no intersection between different clusters, which means that the interval-label classification can learn a better feature representation than the traditional classification. We speculate that the intersection of features of different categories for traditional classification tasks is that when the traditional classification task minimizes the empirical risk, it also pays attention to the non-semantic features, while the values of non-semantic features are all the same for MNIST. Furthermore, non-semantic (non-robust) features cause the vulnerability of the deep learning system (Ilyas et al., 2019).

### 3.3. Robustness Analysis

In this section, we experiment to explore the robustness of the interval-label classification task on both white-box and black-box scenarios. We implement the traditional and interval-label classification tasks on the same deep neural network, only the last fully-connected layer being different. The corresponding experimental results are shown in Tables 1 and 2. For all scenarios, the interval length $\beta$ is set as 16, while the length of the gap between two adjacent interval labels $\alpha$ is set as 4. TRA, HIN, and INT represent the

Table 1. Comparison of attack success rates between traditional and interval-label classification tasks in the white-box attack scenario with different attack intensities on MNIST.

| Model | Task | FGSM | | | PGD | | | LEG |
|---|---|---|---|---|---|---|---|---|
| | | 0.1 | 0.2 | 0.3 | 0.1 | 0.2 | 0.3 | |
| NN | TRA | 0.598 | 0.792 | 0.858 | 0.666 | 0.971 | 0.998 | 0.983 |
| | HIN | 0.275 | 0.349 | 0.378 | 0.290 | 0.491 | 0.597 | 0.984 |
| | INT | 0.000 | 0.000 | 0.001 | 0.001 | 0.005 | 0.015 | 0.985 |
| AlexNet | TRA | 0.866 | 0.972 | 0.984 | 0.937 | 0.998 | 1.000 | 0.982 |
| | HIN | 0.959 | 0.998 | 0.999 | 0.953 | 0.997 | 0.998 | 0.981 |
| | INT | 0.003 | 0.002 | 0.003 | 0.003 | 0.005 | 0.016 | 0.978 |
| ResNet | TRA | 0.660 | 0.861 | 0.912 | 0.991 | 1.000 | 1.000 | 0.985 |
| | HIN | 0.108 | 0.161 | 0.237 | 0.210 | 0.351 | 0.534 | 0.984 |
| | INT | **0.010** | **0.009** | **0.010** | **0.009** | **0.013** | **0.010** | **0.990** |

Table 2. Comparison of attack success rates between traditional and interval-label classification tasks in the white-box attack scenario with different attack intensities on ImageNet.

| Model | Task | FGSM | | | PGD | | | LEG |
|---|---|---|---|---|---|---|---|---|
| | | 3/255 | 6/255 | 9/255 | 3/255 | 6/255 | 9/255 | |
| AlexNet | TRA | 0.405 | 0.550 | 0.663 | 0.921 | 0.989 | 0.997 | 0.747 |
| | HIN | 0.223 | 0.405 | 0.562 | 0.984 | 0.987 | 0.996 | 0.747 |
| | INT | 0.127 | 0.134 | 0.134 | 0.763 | 0.771 | 0.782 | 0.782 |
| DenseNet | TRA | 0.318 | 0.460 | 0.577 | 0.991 | 0.999 | 1.000 | **0.795** |
| | HIN | 0.508 | 0.810 | 0.920 | 0.989 | 0.995 | 0.998 | 0.793 |
| | INT | 0.064 | 0.063 | 0.058 | **0.648** | **0.657** | **0.646** | 0.769 |
| ResNet | TRA | 0.290 | 0.446 | 0.624 | 0.824 | 0.957 | 0.994 | 0.777 |
| | HIN | 0.248 | 0.480 | 0.613 | 0.821 | 0.937 | 0.988 | 0.775 |
| | INT | **0.049** | **0.047** | **0.047** | 0.659 | 0.663 | 0.673 | 0.768 |

traditional classification task, traditional classification task with hinge loss, and the interval-label classification task, respectively. LEG represents the accuracy on the test set.

The experimental results demonstrate that the interval-label classification task is much more robust than the traditional classification task in most cases. In the traditional classification task, the robustness is at odds with the accuracy (Tsipras et al., 2019). However, according to Tables 1 and 2, it is noted that, as to the interval-label classification task, the higher test accuracy, the more robust it is. The reason is that when an example is correctly classified, its loss will be 0. So it is difficult for the adversary to explore an efficient gradient to craft adversarial examples. So we conclude that interval-label classification can alleviate the competition between robustness and accuracy.

### 3.4. Adversarial Transferability between Interval-label Classification Tasks with Different $\alpha$ and $\beta$

In this subsection, we experiment to investigate the adversarial transferability of PGD adversarial examples among interval-label classification tasks with different $\alpha$

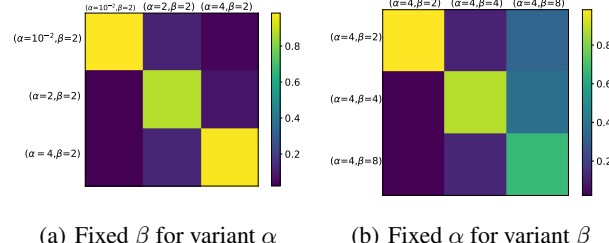

(a) Fixed $\beta$ for variant $\alpha$      (b) Fixed $\alpha$ for variant $\beta$

Figure 3. Adversarial transferability between different interval-label classification tasks under PGD attack, and the attack intensity is set as 0.01. The transferability of adversarial examples is asymmetry for different tasks. So the confusion matrix is not symmetrical. The row task is the threat task, while the column task is the victim task.

and $\beta$ on ImageNet. $\eta$ is the attack intensity of the PGD attack. All the interval-label classification tasks are defined on the ResNet18. In Fig. 3, the lighter color represents a higher attack success rate. We consider two scenarios, namely, the length of the interval label is fixed, and the length of the gap between two adjacent interval label is fixed.

According to Fig. 3, we know that bigger $\alpha$ and $\beta$ will increase the robustness of interval-label classification. Moreover, we also note that even though the white-box attack success rate is high in some cases, the adversarial example is a little hard to transfer between the interval-label tasks with different $\alpha$ and $\beta$. The reason is that when $\alpha$ and $\beta$ are small, the interval-label classification is approximate to the traditional classification. In a real-world application, a deep neural network may be deployed in variant devices. So if an adversary has successfully attacked a device, the crated adversarial example may have a chance to attack the rest devices. The interval-label classification task promises to address the issue mentioned above since the defender can adopt a different combination of $\alpha$ and $\beta$.

## 4. Conclusion

In this paper, we extend the definition of the adversarial robustness of DNNs to the deep learning system. Furthermore, we first investigate the learning task robustness of the deep learning system. The experimental results show that the interval-label classification task can alleviate the competition between accuracy and robustness for the deep learning system. Moreover, the traditional classification task is easily transformed into the interval-label classification task. However, the interval-label classification task converges slower than the traditional classification task. In the future, a more efficient interval-label classification deep learning system will be explored.

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
