# OpenReview forum: "Is It Time to Redefine the Classification Task for Deep Learning Systems?"
_ICML.cc/2021/Workshop/AML — ICML 2021 Workshop AML Oral_

### Official Review · Reviewer_amTX · 2021-06-20
**A novel and inspiring paper to discuss the robustness of predefined learning tasks.**

**Rating:** Accept
**Confidence:** 4

**Review:**

The paper is well organized and clear.

It is a novel work which extends the robustness from model to task. It proposes a transformation to convert a classification task into an interval prediction task and its inverse transformation.

The experiments are exhausted to demonstrate the robustness of the interval prediction task compared to corresponding classification task.

This is a significant discovery in robustness community which inspires further researches on the robustness of other tasks by converting predefined tasks.

I'm confused by the sentence “we can randomly assign s 0 , α, and β for each interval label.” in Line 109. It needs more explanation.

The font size in Figure 3 is too small.

---

### Decision · Program_Chairs · 2021-06-21

**Decision:**

Accept (Oral)

**Comment:**

The reviewer acknowledged the novelty and clarity of this paper. The paper is well organized and the experiments are exhausted.